# Sox9 Inhibits Cochlear Hair Cell Fate by Upregulating Hey1 and HeyL Antagonists of Atoh1

**DOI:** 10.3390/cells12172148

**Published:** 2023-08-25

**Authors:** Mona Veithen, Aurélia Huyghe, Priscilla Van Den Ackerveken, So-ichiro Fukada, Hiroki Kokubo, Ingrid Breuskin, Laurent Nguyen, Laurence Delacroix, Brigitte Malgrange

**Affiliations:** 1Laboratory of Developmental Neurobiology, GIGA-Neurosciences, University of Liege, 4000 Liege, Belgium; mona.veithen@uliege.be (M.V.); aurelia.huyghe@lyon.unicancer.fr (A.H.); p.vandenackerveken@outlook.be (P.V.D.A.); ingrid.breuskin@gustaveroussy.fr (I.B.); ldelacroix@uliege.be (L.D.); 2Laboratory of Stem Cell Regeneration and Adaptation, Graduate School of Pharmaceutical Sciences, Osaka University, Osaka 565-0871, Japan; fukada@phs.osaka-u.ac.jp; 3Graduate School of Biomedical and Health Sciences, 1-2-3 Kasumi, Minamiku, Hiroshima 734-8551, Japan; hkokubo@hiroshima-u.ac.jp; 4Laboratory of Molecular Regulation of Neurogenesis, GIGA-Neurosciences, University of Liege, 4000 Liege, Belgium; lnguyen@uliege.be

**Keywords:** Sox9, cochlea, organ of Corti, transfection, differentiation, development

## Abstract

It is widely accepted that cell fate determination in the cochlea is tightly controlled by different transcription factors (TFs) that remain to be fully defined. Here, we show that Sox9, initially expressed in the entire sensory epithelium of the cochlea, progressively disappears from differentiating hair cells (HCs) and is finally restricted to supporting cells (SCs). By performing ex vivo electroporation of E13.5–E14.5 cochleae, we demonstrate that maintenance of Sox9 expression in the progenitors committed to HC fate blocks their differentiation, even if co-expressed with Atoh1, a transcription factor necessary and sufficient to form HC. Sox9 inhibits Atoh1 transcriptional activity by upregulating Hey1 and HeyL antagonists, and genetic ablation of these genes induces extra HCs along the cochlea. Although Sox9 suppression from sensory progenitors ex vivo leads to a modest increase in the number of HCs, it is not sufficient in vivo to induce supernumerary HC production in an inducible Sox9 knockout model. Taken together, these data show that Sox9 is downregulated from nascent HCs to allow the unfolding of their differentiation program. This may be critical for future strategies to promote fully mature HC formation in regeneration approaches.

## 1. Introduction

The mammalian cochlea has a specialized sensory epithelium, the organ of Corti, which is involved in sound perception. Cochlear sensory epithelium morphogenesis is a highly complex and stereotyped process that gives rise to a specific mosaic pattern of hair cells (HCs) and supporting cells (SCs). In mice, the development of the organ of Corti from the otocyst can be divided in two phases [1]. Starting around embryonic stage 12.5 (E12.5), the first step specifies a prosensory area within the ventral part of the developing otocyst. During the second phase, beginning at E14.5–E15.5 at the base of the cochlea, the HCs and their accompanying SCs differentiate from common progenitor cells within the prosensory epithelial domain.

Prosensory progenitors exit the cell cycle around E12.5 and form a post-mitotic zone of non-proliferating cells [2]. In contrast to fish and birds, mature cells of the mammalian organ of Corti do not contribute to cellular regeneration after injury, as they cannot re-enter the cell cycle. Unsurprisingly, hair cell loss in the adult mammalian auditory system is irreversible and is a leading cause of permanent hearing loss. Therefore, defining the transcriptional programs that pattern the cochlea and specify HC and SC fate from progenitors will likely contribute to establishing new strategies for treating neurosensory deafness [3,4].

Previously, several studies have brought to light the mechanisms contributing to cochlear terminal cell differentiation (after E14.5), and major players, such as the transcription factor (TF) Atoh1 and the Notch signaling pathway, have been identified. The proneural basic helix–loop–helix (bHLH) *Atoh1* gene has been identified as the earliest specific gene required for HC differentiation. Deletion of *Atoh1* leads to a complete absence of HCs, whereas Atoh1 overexpression induces ectopic HCs in both sensory and non-sensory regions of the cochlea [5,6,7]. Differentiating HCs express Delta1 and Jagged2 ligands [8,9], which signal to Notch-expressing neighboring SCs to inhibit them from acquiring HC fate, in a process named lateral inhibition [10]. Conditional deletion of either *Jagged2* or *Delta1* in the cochlea leads to an overproduction of HCs, via a fate switch of SCs [11,12]. Upon activation through ligand binding, the Notch receptor is cleaved and its intracellular domain (NICD) associates with RBP-Jκ in the nucleus to transactivate target genes [13], including the Hairy Enhancer of Split (Hes) family members, Hes1 and Hes5. In the cochlea, these transcriptional repressors act as critical regulators of cell fate by inhibiting *Atoh1* transcription [14], and deletion of either of these genes induces supernumerary HCs in the cochlea [15].

Sox9, a high-mobility group (HMG) box TF, is expressed during embryonic development in several tissues including chondrocytes, lung, pancreas, heart, the central nervous system, and the inner ear [16,17,18]. Typically, Sox9 is essential during early stages of development to maintain a progenitor cell pool, while, at later stages of organogenesis, it plays a role in terminal cell differentiation [19,20,21]. In humans, mutations of *Sox9* are associated with inherited genetic birth defects, including campomelic dysplasia (CD), an autosomal dominant disease characterized by skeleton malformation, XY sex reversal, and a high neonatal lethality rate [22]. Surviving patients often suffer sensorineural hearing loss associated with a malformation of the inner ear canals [22,23,24]. Because Sox9 is one of the first specific markers of the otic placode, various studies have focused on the early functions of Sox9 in the inner ear. In Xenopus [25,26] and Zebrafish [27], loss of *Sox9* results in the failure of the otic placode and vesicle development, while in mice, it is required for placode invagination [28].

Given the known role of Sox9 in cell fate specification in multiple organs [29] and its expression in the mammalian cochlea [18], we investigated the contribution of Sox9 to mouse cochlear differentiation. Gain of function experiments revealed that Sox9 prevents HC differentiation by reducing Atoh1 activity. This effect was mediated by upregulating Hey1 and HeyL factors, which in turn inhibit Atoh1 transactivation potential. Altogether, our data support that Sox9 is a regulator of cell fate in the sensory epithelium of the organ of Corti.

## 2. Materials and Methods

### 2.1. Mouse Strains and Tamoxifen Treatment

Hey1^+/−^, HeyL^+/−^ [30], Sox9^eGFP^ [31], Sox2^CreERT2^ [32], Sox9^lox/lox^ [33]**,** and wild-type NMRI or BALB/c mice (Animal Facility of the University of Liège) were used. A Cre-inducible R26R^EYFP^ responder strain [34] was also incorporated into the breeding scheme to monitor the extent of Cre-mediated recombination.

Mice were time-mated, and conception was confirmed by the presence of a vaginal plug, which was considered as day 0.5 (E0.5). The day of birth was regarded as P0. All experimental procedures and protocols were reviewed and approved by the Institutional Animal Care and Use Ethics Committee of the University of Liège (Belgium). The “Guide for the Care and Use of Laboratory Animals”, prepared by the Institute of Laboratory Animal Resources, National Research Council, and published by the National Academy Press, was followed carefully as well as European and local legislation.

Tamoxifen was dissolved at 30 mg/mL in a sunflower oil/ethanol mixture (9:1) and administered to time-pregnant mice orally (100 µL) and by intraperitoneal injection (100 µL) for 2 consecutive days (E12.5–E13.5).

### 2.2. Plasmid Constructions

Plasmids encoding specific shRNAs were constructed by cloning synthesized oligonucleotides into pCA-b-EGFPm5 silencer 3 vector [35] with the following targeting sequences: 5′-CAGACTCACATCTCTCCTAAT-3′ (shRNA-Sox9#1), 5′-CTCCACCTTCACTTACATGAA-3′ (shRNA-Sox9#2), 5′-AGACCGAATCAATAACAGTTT-3′ (shRNA-Hey1) and 5′-GCTGTTGACTTCCGGAGTAT-3′ (shRNA-HeyL). A control shRNA vector was generated by cloning a sequence with no significant homology to any known gene sequence from a mouse: 5′-TACGCGCATAAGATTAGGG-3′.

Expression plasmids were constructed by cloning Atoh1, Sox9, Hey1, and HeyL coding sequences into the bicistronic pCAGGS-IRES-GFP vector. Using Expand™ high-fidelity PCR system (Merck Life Science BV, Hoeilaart, Belgium), we generated Sox9 coding sequence corresponding to amino acids 1–304 and cloned it in the same plasmid to generate a DN-Sox9 expression vector.

Luciferase reporters controlled by Atoh1 (7EBox-Luc) and Sox9 binding sites (called pCol2a1-Luc in this paper) as well as those driven by promoters of *Sox9* (pSox9-luc), *Hey1* (−604/+87pHey1-Luc), and *HeyL* (1168/+288pHeyL-Luc) were described previously [36,37,38,39,40]. Additional reporter vectors were created by inserting shorter promoter fragments into the pLUC reporter construct. Inserts were generated by high-fidelity PCR amplification using forward primers containing NheI site and reverse primers containing XhoI site (Appendix A), except for pHey1 (−50/+3), which was synthesized (IDT, Leuven, Belgium). All constructs were verified by sequencing.

### 2.3. Cell Culture and Transfection

Immortalized organ of Corti cells derived from the mouse at E13.5, UB/OC-1 cells [41], were obtained from Pr. Matthew Holley (Department of Biomedical Science, Sheffield, UK) and cultured in MEM (Lonza, Basel, Switzerland) supplemented with 10% fetal bovine serum (Lonza) and 50 U/mL γ-Interferon (Sigma) in a humidified 5% CO_2_ atmosphere at 33 °C. HEK293 cells were maintained in DMEM medium supplemented with 10% FBS in a humidified 5% CO_2_ atmosphere at 37 °C.

Cells were transfected in 24-well plates with a total of 500 ng DNA and 1–1.5 µL of Lipofectamine 2000 reagent (Invitrogen, Carlsbad, CA, USA) 48 h before experimentation.

Validated endoribonuclease-prepared siRNA for silencing *Hey1* (N°EMU044841 directed against the exon 5 of mouse Hey1) and *HeyL* (N°SASI_Mm01_00137911) genes were purchased from Merck Life Science BV (Hoeilaart, Belgium). In transfection experiments combining DNA plasmids and siRNAs, the cells were first transfected with 20 nM siRNA using Lipofectamine 2000 and further transfected two days later with 20 nM siRNA and 500 ng DNA.

### 2.4. Western Blot

Cells were lysed on ice in a solution containing 50 mM Tris HCl, pH7.4, 1% Triton X-100, 150 mM NaCl, 10 mM NaF, 1 mM Na3VO4, and protease inhibitors (Protease Inhibitor Cocktail Tablets, Roche). Protein concentration was assessed using the Bradford method. Protein lysates (30 µg) were separated by 10% SDS-polyacrylamide gel electrophoresis and transferred onto PVDF membranes (Millipore, Burlington, MA, USA). Non-specific binding was blocked with 5% dried fat-free milk in TTBS (50 mM Tris-HCl pH 7.4, 150 mM NaCl, 0.1% Tween 20) for 1 h at room temperature and subsequently incubated overnight at 4 °C with the following primary antibodies: rabbit polyclonal antibody to Sox9 (1:1000, Santacruz, Santa Cruz, CA, USA) and mouse monoclonal antibody to β-actin (1:5000, clone AC-15, Sigma Aldrich). After washing steps in TTBS, membranes were incubated with the appropriate horseradish peroxidase-conjugated secondary antibody (1/5000, AbCam, Cambridge, UK) for 2 h at room temperature. Protein expression was detected by enhanced chemiluminescence (ECL, GE Healthcare, Chicago, IL, USA).

### 2.5. RT-qPCR

Total RNA from UB/OC1 cells or dissected cochleae was extracted using TriPure Isolation Reagent (Roche) with a DNase I (Roche) step according to the manufacturer’s instructions. cDNA was prepared from 1 µg of RNA using the RevertAid H Minus First Strand cDNA Synthesis Kit (Fisher scientific BVBA, Brussels, Belgium) and used for qPCR in a LightCycler 480 (Roche, Machelen, Belgium). Expression levels were normalized to the expression of *glyceraldehyde-3-phosphate dehydrogenase* (GAPDH). All amplifications were performed in duplicate, and at least three biological replicates were performed. PCR primer sequences that were used are listed in Appendix A.

### 2.6. Luciferase Assay

UB/OC1 or HEK293 cells (5 × 10^4^ cells/well in 24-well plates) were transfected using 200 ng reporter vector (Firefly Luciferase), 10 ng pRL-SV (Renilla Luciferase control vector, Promega, Madison, WI, USA), and 300 ng expression vectors (when necessary). The total amount of DNA was kept constant by adding the control pCAGGS plasmid. Cell lysates were assayed for luciferase activity two days post-transfection using the Promega Dual Luciferase kit and a Berthold LB 960 microplate luminometer. Firefly Luciferase activity was reported to Renilla Luciferase activity to normalize for the differences in transfection efficiencies. Each experiment was repeated at least three times, and each measurement was performed in duplicates.

### 2.7. Organotypic Cochlear Cultures

#### 2.7.1. Tissue Isolation and Culture

Cochleae of stage E13.5 or E14.5 were collected in PBS (Lonza). Each organ of Corti was freed from surrounding tissues and explanted intact onto the surface of a sterile membrane (Millicell^®^, Millipore) in DMEM (Gibco, Billings, MT, USA) containing N1 supplement (Invitrogen), glucose 0.15%, penicillin (100 U/mL), and insulin (5 µg/mL). Cultures were maintained in a 5% CO_2_-humidified incubator at 37 °C.

#### 2.7.2. Electroporation

Cochlear ducts were injected with a 2 µg/µL DNA solution in 0.5% Fast Green (Sigma) and placed in a Sonidel electroporation chamber (CUY520P5). A total of 8 electrical pulses were applied at 30 V (50 ms duration) at 800 ms intervals using a square wave electroporation system (ECM-830 BTX, San Diego, CA, USA). Cochleae were dissected after electroporation and cultured on Millicell^®^ for 6 days to allow terminal cell differentiation. The culture medium was replaced after the first 3 days.

### 2.8. Immunohistochemistry

For immunohistological analysis, heads were fixed with 4% paraformaldehyde (Sigma-Aldrich, Burlington, MA, USA) for 4–6 h at 4 °C. After three PBS rinses, the fixed heads were immersed in 20% sucrose in PBS overnight at 4 °C. Tissues were frozen, embedded in 7.5% gelatin/15% sucrose in PBS, then sectioned at 12 µm, mounted on SuperFrost^®^ slides, and stored at −80 °C. Sections were washed three times with PBS and blocked for 1 h with 0.25% gelatin and 0.1% Triton X100 (Sigma-Aldrich) in PBS at room temperature. For cochlear explants, the tissues were fixed in 4% paraformaldehyde for 10 min at room temperature, washed in PBS, and then directly processed for blocking. Antigens were unmasked, when necessary, by incubating samples in an antigen retrieval solution (Dako) for 15 min at 95 °C, followed by slow cooling back to room temperature before blocking. Primary antibodies were incubated overnight at 4 °C in the same blocking solution. The following primary antibodies were used in various combinations: Myosin 6 (Myo6, rabbit, 1:250, Santa Cruz); Parvalbumin (mouse, 1:250, Sigma-Aldrich); Prox1 (rabbit, 1:500, Millipore); Sox2 (goat, 1:200, Santa Cruz); Sox9 (rabbit, 1:100, Millipore or gift from Brigitte Boizet); and GFP (chicken, 1:500, Aves Labs). Samples were subsequently washed three times in PBS and incubated for 45 min at room temperature in a blocking solution containing a secondary antibody (1:1000) conjugated to FITC-, RRX-, or Cy5 fluorophores (Jackson Immunoresearch Laboratories, West Grove, PA, USA). After three rinses in PBS, preparations were mounted in Vectashield containing DAPI (Hard Set Mounting Medium, Vector Laboratories, Burlingame, CA, USA) and examined with a confocal microscope (NIKON A1, Brussels, Belgium).

### 2.9. Quantification of HC Fate in Cultured Explants

Immunostained explants were analyzed with a NIKONA1 confocal microscope, with a 60× magnification, and images were taken at different focal planes spanning the depth of the sensory epithelium (hair cells and underlying supporting cells). Except for the apical region of the explant, all GFP-positive cells residing in the region comprising the hair cells and the supporting cells displaying strong Sox2 labelling (from inner phalangeal cells to Deiters cells) were considered for further analysis. Using the cell counter tool in ImageJ software (version 1.51h), cells expressing Myo6 and no or low Sox2 were considered as hair cells, and their proportion within all GFP-positive cells was determined manually for each explant. Ectopic HCs, which are only induced upon Atoh1 transfection, were determined by analyzing GFP-positive cells that are present in Kölliker’s organ, the region located at the medial side of inner HCs. All data were collected from a minimum of 5 explants (see Appendix A), from 2–3 independent cultures, and the results are expressed as the percentage mean ± SEM.

### 2.10. Supernumerary HC Counts

Whole-mount cochlear samples from newborn mice were dissected, immunostained for HC marker Myo6, and imaged with a NIKONA1 confocal microscope. Extra HCs were quantified all along the cochlea except for the most apical part. The length of the sensory epithelium submitted to analysis was measured using ImageJ software, and the results are expressed as the mean ± SEM of the number of supernumerary HCs per mm of cochlea (more than *n* = 4 for each group).

### 2.11. In Situ Hybridization

E14.5, E17.5, or P0 heads were fixed in 4% paraformaldehyde in PBS overnight at 4 °C, sunk in 20% sucrose in PBS overnight at 4 °C, and embedded in 7.5% gelatin with 15% sucrose in PBS. Cochlear explants were fixed in 4% paraformaldehyde for 4 h at room temperature and washed in PBS before ISH. ISH was performed on 12 µm thick transverse sections of the head or entire cochleae using digoxigenin-labelled riboprobes, as previously described [42]. Plasmids containing full-length mouse Hey1 and HeyL cDNAs were provided by Manfred Gessler.

### 2.12. Statistical Analysis

Statistical analyses were carried out using the unpaired Student’s *t*-test for pairwise comparisons or a one-way ANOVA followed by Tukey’s post hoc test using GraphPad Prism 8 (GraphPad Software, Boston, MA, USA) for all the other studies. The results are presented as mean ± SEM and were considered significant when *p* < 0.05. The number of biological replicates relevant for individual experiments are stated in the Results section or Appendix A.

## 3. Results

### 3.1. Analysis of Sox9 Expression Pattern in the Developing Mouse Cochlea

Previous analyses have shown that Sox9 is expressed at E8.5 in the nascent otic placode and at E9.5 in the entire otic vesicle [18,28]. We performed in situ hybridization on E13.5 cochlear sections to extend this analysis to later developmental stages. We found that Sox9 mRNA is highly expressed in the otic epithelium throughout the cochlear duct and in the otic capsule (Figure 1A). Consistent with published results by Mak et al. [18], Sox9 protein was detected by immunohistochemistry in the entire cochlear epithelium (Figure 1B). This pattern of expression was further confirmed in the developing cochleae of a transgenic mouse line in which GFP expression is under the control of the Sox9 regulatory sequences [31] (Figure 1C, Sox9^EGFP^). Between E14.5 and E18.5, Sox2-positive progenitor cells are subjected to a baso-apical wave of differentiation generating the HC and their surrounding SC [1]. At E16.5, Sox9 immunoreactivity was lost specifically in some Sox2-expressing cells that likely correspond to differentiating HC in the basal turn of the cochlea. At the same time, it was still detected in all prosensory cells of the apical turn (compare Figure 1G–I to Figure 1D–F). Finally, Sox9 remains absent from Myo6-positive HC at P0 and is restricted to the SC in the organ of Corti (Figure 1J–L).

### 3.2. Sox9 Inhibits Cochlear Hair Cell Fate Ex Vivo

We next sought to characterize the functional significance of Sox9 downregulation from the nascent HC population of the developing cochlea. We performed Sox9 gain-of-function experiments to test whether we could change cell fate in the E14.5 organs of Corti. Six days after electroporation, when terminal differentiation was complete, cultured explants were immunolabelled with HC and SC markers. In the sensory epithelium, 28.5 ± 2.15% of cells transfected with the control GFP plasmid became fully differentiated HCs and were positive for Myo6 or Parvalbumin (Figure 2B,C,P and Appendix A). In contrast, Sox9-GFP electroporated cells rarely expressed Myo6 or Parvalbumin (1.52% ± 1.13, Figure 2D–G,P and Appendix A). Within the HC layer, some of these Myo6-negative Sox9 transfected cells still showed weak staining for Sox2 (Appendix A, white arrows), similar to that observed in the neighboring non-transfected HCs. However, for most of them, Sox2 was absent from the nucleus (Appendix A, yellow arrowheads). These cells did not express SC-specific markers Prox1 or p27, indicating that HC fate was inhibited but not at the profit of a switch towards SC fate. Within the SC layer, transfected cells do not seem to be affected by Sox9 overexpression, as they expressed normal levels of Prox1, Sox2, or p27 (Figure 2I–K and Appendix A).

To further study the role of Sox9 in cochlear cell fate and patterning, we performed loss-of-function experiments. Two shRNAs targeting different regions of Sox9 transcript were tested in UB/OC1 cells, which derive from a murine otocyst and are classically used as a model for organ of Corti cell differentiation [41]. We transfected them in an otocyst-derived UB/OC1 cell line to establish their efficacy and performed a Western blot for Sox9 48 h later. Both shRNAs efficiently reduced the expression of Sox9 (Appendix A). We then performed a knockdown of Sox9 by electroporating these shRNAs in E13.5 cochlear explants and analyzed the phenotype of transfected cells after 6 days in vitro (DIV). A modest increase in the proportion of Myo6-GFP-positive cells was observed in the presence of both Sox9 shRNAs (Figure 2M–O and Appendix A), revealing the formation of extra HCs.

Altogether, these results suggest that Sox9 inhibits HC differentiation from progenitors and that the correct cell patterning in the organ of Corti requires the downregulation of Sox9 in nascent HCs.

### 3.3. Sox9 Inhibits Atoh1 Activity

The specification of HCs requires the transcriptional factor of Atoh1 [5,7,8]. Atoh1 expression is first detected at a low level in progenitor cells of the organ of Corti, before cell differentiation, and then further upregulated in differentiating HCs [43,44,45]. Because we showed that modulating Sox9 expression in situ interferes with HC differentiation, we investigated the impact of Sox9 overexpression on Atoh1 function. Ex vivo electroporations of Atoh1-GFP alone or combined with Sox9 in E14.5 cochlear explants were performed and subsequently analyzed after 6 DIV. Consistent with a previous report [7], most Atoh1-GFP-transfected cells expressed the HC marker Parvalbumin within the organ of Corti (86.42% ± 3.65, Figure 3A,B,E). Interestingly, the number of transfected cells adopting HC fate was drastically reduced upon co-transfection with Sox9-GFP (10.25%± 3.17, Figure 3C–E), indicating that Sox9 is able to counteract Atoh1-mediated HC induction. Noteworthy is that Atoh1 transfection in cells residing in Kölliker’s organ gave rise to ectopic HCs. Indeed, the vast majority of GFP-positive cells located on the medial side of the HC region expressed Parvalbumin (Figure 3A,B) and KO. The ability of Atoh1 to induce HC fate outside the organ of Corti was also impaired by the presence of Sox9-GFP, as many cotransfected cells lack the Parvalbumin marker (Figure 3C,D, white arrows). Sox9 could act by inhibiting Atoh1 DNA binding or transactivation of specific target genes, thus blocking the initiation of the HC developmental program. Therefore, we monitored Atoh1 transcriptional activity by performing in vitro reporter assays using a multimerized Atoh1 responsive element (seven repeats of Ebox; see Figure 3F) upstream of the luciferase coding sequence [14]. This analysis was performed in the UB/OC1 cell line. When Sox9 was co-transfected with Atoh1, the reporter activity was significantly reduced dose-dependently, while it had no effect in the absence of Atoh1 (Figure 3H). These results indicate that Sox9 interferes with the transcriptional activity of Atoh1.

To decipher whether the effect of Sox9 involves the upregulation of Atoh1 modulators, we decided to block Sox9 transcriptional activity by using a dominant negative form of Sox9 (DN-Sox9). This mutant lacking the C-terminal transactivation domain represses the Sox9-dependent activation of one of its main targets, i.e., the Col2a1 promoter (Appendix A) [46]. In the Atoh1 reporter assay (Figure 3H), DN-Sox9 suppressed, in a dose-dependent manner, the inhibitory effect of Sox9 on Atoh1. Accordingly, DN-Sox9 also suppresses, dose-dependently, the inhibitory effect of Sox9 on HC fate when co-electroporated with Sox9 in cochlear explants (Appendix A).

Collectively, these results indicate that Sox9 activates the transcription of genes that inhibit Atoh1 transactivation potential and HC induction.

### 3.4. Sox9 Activates Hey1 and HeyL Gene Expression

To identify Sox9 target genes acting as Atoh1 putative antagonists, we examined the role of Hes/Hey and Id proteins, two families of negative regulators of bHLH proteins [47]. Interestingly, some of these factors are implicated in cochlear differentiation [48,49]. We transfected a Sox9 expression plasmid in UB/OC1 cells (an otic cell line) and measured the relative mRNA expression level of Hes1, Hes5, Hey1, Hey2, HeyL, and Id1-3 by RT-qPCR. Transcripts for all genes were detected at low levels in basal conditions except for Hes5, which was not detected. We found that the expression of Hey1 and HeyL was strongly enhanced by Sox9 overexpression in the otic cell line (Figure 4A), and this was prevented by co-transfecting the dominant negative DN-Sox9 (Appendix A).

We next performed in situ hybridization analyses at E14.5, E16.5, and P1 to examine the spatio-temporal expression patterns of Hey1 and HeyL in developing organs of Corti (Appendix A). These data revealed that Hey1 and HeyL expression profiles were consistent with the Sox9 expression pattern and Atoh1 modulation in the developing organ of Corti. Indeed, as HC differentiation arises in the murine organ of Corti, between E14.5 and E16.5, the expression of Hey1 and HeyL is progressively restricted to future SCs, in agreement with previously published data [49,50].

To confirm Sox9 regulation of Hey1 and HeyL expression in the cochlea, we first electroporated Sox9 in the E14.5 organ of Corti explants and performed in situ hybridization to detect Hey1 and HeyL expression after 6 DIV. Hey1 and HeyL mRNAs were specifically detected along the length of the sensory epithelium, and GFP-positive cells, overexpressing Sox9, showed increased Hey1 and HeyL expression levels (Figure 4B).

We also performed luciferase assays in UB/OC1 cells to study the effect of Sox9 on Hey1 and HeyL promoter reporters. The murine Hey1 promoter fragment (ranging from −602 to +87 positions, relative to transcription start site TSS) showed increased transcriptional activity upon Sox9 expression in UB/OC1 cells (Figure 4C). This Sox9-dependent increase was also observed for a shorter promoter spanning −95/+3, whereas it was lost on a −50/+3 fragment. These results indicated that the Hey1 promoter includes a Sox9 responsive element located between 90 and 50 bp upstream of the TSS. Sox9 also activated the HeyL promoter when a minimum of 200 bp upstream of the TSS was used (Figure 4D). This response was reduced progressively as promoter fragments were shortened to 150 and 80 bp. These in vitro experiments revealed that Hey1 and HeyL promoters respond to Sox9 through sequences located within their proximal region.

Altogether, these results demonstrate that Hey1 and HeyL are Sox9 target genes activated in the cochlea and suggest that these factors mediate the inhibitory effect of overexpressed Sox9 on Atoh1 protein and HC differentiation.

### 3.5. Overexpression of Hey1 and HeyL Prevents Hair Cell Differentiation

We next assessed whether Hey1 and HeyL could modulate cell fate in the organ of Corti. Firstly, we electroporated Hey1- and HeyL-expression plasmids in the E14.5 organ of Corti explants and harvested them after 6 DIV. Within the HC layer, the majority of cells transfected by Hey1 or HeyL did not express Myo6, indicating that these Hey factors strongly inhibited HC differentiation (Figure 5A–G, and Appendix A). This outcome resembled the effects observed upon Sox9 overexpression. We next examined whether homozygous deletion of Hey1 and HeyL genes leads to a modification of HC number in the developing mouse cochlea. At birth, cochleae from single Hey1 or HeyL knock-out embryos displayed a few extra HCs in the organ of Corti, a phenotype almost never detected in wild-type littermates (Figure 5H–N). The number of supernumerary HCs was further increased when Hey1 and HeyL were absent (Figure 5M,N). Together, these results indicate that both Hey1 and HeyL prevent HC fate acquisition and contribute to the development of a highly ordered mosaic in the organ of Corti.

### 3.6. Sox9 Inhibition of Atoh1 Partially Relies on Hey1 and HeyL Upregulation

We next investigated whether Hey1 and HeyL were required downstream of Sox9 to inhibit Atoh1 activity. We tested this hypothesis by co-transfecting the 7Ebox-Luc reporter, Sox9- and Atoh1-expressing vectors in UB/OC1 cells with or without endoribonuclease-prepared siRNAs targeting Hey1 andHeyL. After 48 h, we assessed the Atoh1-induced luciferase reporter activity (Figure 6A). Our results showed that preventing Hey1 and HeyL expression reduced the inhibitory effect of Sox9 on Atoh1 activity. To confirm these results in the cochlea, we co-electroporated Sox9-GFP plasmid and shRNA directed against Hey1 and HeyL in E14.5 explants. After 6 days in culture, down-regulation of endogenous Hey1 and HeyL significantly reduced the ability of Sox9 to inhibit HC fate (Figure 6B,C, percentage of GFP+Myo6+ cells for Shctrl: 24.26% ± 1.33, Sox9-GFP+Shctrl: 2.13% ± 0.79, and Sox9-GFP+ShHey1/L: 7.79% ± 1.61). Notably, the percentage of HCs amongst GFP-positive cells was still reduced compared to controls, suggesting that Hey1 and HeyL are not the only mediators of Sox9 inhibitory action on HC differentiation.

### 3.7. Sox9 Loss Leads to Supernumerary HCs but Not upon Sox2 Hypomorphism

To study cochlear cell patterning upon depletion of Sox9 in vivo, we crossed Sox9^lox/lox^ mice with Sox2^CreERT2/+^ and R26R^EYFP^ mice. The activity of the Cre recombinase, expressed in the prosensory Sox2-positive cells, was induced by tamoxifen administration at E12.5 and E13.5 prior to terminal cell differentiation and was monitored by visualization of the YFP reporter (Figure 7A). As reported previously, the presence of the Cre allele, substituting the Sox2 gene in the Sox2^CreERT2/+^ mouse line, leads to Sox2 hypomorphism [51], which is accompanied by some overproduction of HCs within the sensory epithelium (Figure 7A,B; compare Sox2^+/+^ and Sox2^CreERT2/+^ animals). We thus analyzed cochlear patterning in tamoxifen-treated littermates of all genotypes but found no effect of Sox9 loss upon Sox2 deficiency. The inhibitory action of Sox9 on Atoh1 and HC differentiation may be masked or prevented [52], as Sox2 was also shown to antagonize Atoh1 function [53]. It is also conceivable that Sox2 hypomorphism leads to gene expression changes of additional developmental regulators that compensate for the loss of Sox9 in the Sox2^CreERT2/+^ mouse model.

As the ex vivo knockdown of Sox9 by shRNA promoted the occurrence of supernumerary HCs, we sought to use the Sox9^lox/lox^ model to confirm these findings. The inner ears were electroporated at E13.5 with a Cre-GFP plasmid to delete Sox9, and the proportion of transfected cells that differentiated into HCs was evaluated and compared to GFP-transfected controls. Again, HC fate was significantly promoted in Sox9-depleted cells (Figure 7C,D), indicating that Sox9 fine-tunes the transcriptional program during sensory cell development in the cochlea.

## 4. Discussion

Hearing depends on sound wave detection by the mechanosensory HCs in the cochlea. These cells are highly vulnerable to environmental insults as they are lost following noise or ototoxic drug exposure. In mammals, these cells cannot regenerate, or with poor efficiency and only during the perinatal development period [54]. For many years, researchers have been investigating strategies to promote HC regeneration to replace damaged or lost auditory cells. Because HCs and SCs share common precursors, deciphering the mechanisms that regulate cell fate decisions and early differentiation during inner ear development is necessary to enable attempts to promote HC regeneration.

In this study, we report a novel requirement for Sox9 in fine-tuning the organ of Corti cell patterning. We show that dynamic changes in Sox9 expression during cochlear development are required for terminal HC differentiation from the prosensory cells. We show that Sox9 acts as an inhibitor of Atoh1, the primary inducer of HC development. Its expression must be turned off specifically in cells committed to HC fate to ensure proper differentiation into mature sensory cells.

### 4.1. Sox9 Cell Fate Determiner Is an Inhibitor of Atoh1 in the Cochlea

Atoh1 has been identified as the earliest gene necessary and sufficient for HC development. Its absence results in a complete loss of HCs [5], while its expression is sufficient to induce ectopic HCs [52]. Our results indicate that overexpression of Sox9 in the cochlear prosensory progenitors prevents Atoh1 from inducing HC differentiation and is associated with reduced Atoh1 transcriptional activity. Indeed, in the presence of Sox9, the ability of Atoh1 to activate transcription from Ebox-containing regulatory regions is strongly reduced. As Atoh1 also relies on Ebox enhancers for its regulation [55], the transcription factor is part of a positive feedback loop reinforcing its own expression. As such, expression of Sox9 in nascent HCs reduces Atoh1 transcriptional activity and consequently downregulates Atoh1 and its downstream target genes, thereby preventing the HC differentiation program.

Importantly, blockade of HC differentiation by Sox9 did not result in their conversion towards an SC fate. This contradicts the prevailing view that SC differentiation occurs as a default mode ref. Although SC formation has been extensively studied, and despite the advances in their molecular profile provided by single-cell transcriptomic studies throughout developmental stages, specific regulators of SC fate still need to be identified.

Sox9 overexpression in developing HCs precludes the execution of Atoh1 transcriptional program, but it is not clear what is the outcome of these cells. Previous studies in the nervous system suggest that Sox9 could be an epithelial–mesenchymal transition (EMT) modulator [56,57]. Thus, Sox9-overexpressing nascent HCs could mis-differentiate into a mesoderm-like phenotype.

### 4.2. Functional Redundancy for Sox Factors in the Differentiating Organ of Corti?

Previous reports have suggested similar roles for Sox2, which is also initially expressed in all prosensory cells before becoming restricted to SCs, and that was shown to antagonize Atoh1 function [53]. However, Sox2 is retained in nascent HCs already devoid of Sox9 and is progressively downregulated later on [8,15,18,58]. Expression analyses in human cochleae confirm that Sox9 downregulation coincides with HC differentiation and occurs several weeks before the downregulation of Sox2 [59]. Moreover, Sox2 seems to exert opposite effects during terminal cell differentiation in the developing organ of Corti. It was also shown to increase the transcription of Atoh1, favoring HC differentiation when transfected with Eya1/Six1 factors [60]. Similarly, Sox2 can induce Atoh1 and form ectopic HCs in the chick [61].

Altogether, both Sox2 and Sox9 are involved in the intricate regulatory networks that control cell differentiation within the organ of Corti, but they seem to have distinct functions.

### 4.3. Hey1 and HeyL Are Downstream Effectors of Sox9

Our data indicate that Sox9 regulates cochlear cell differentiation in a Hey1/HeyL-dependent manner. Interestingly, mice lacking both Hey1 and HeyL have an increased number of HCs. However, this relatively mild effect may be due to a possible compensatory effect by other members of the Hes/Hey gene family. Indeed, Hes1, Hes5, or Hey2 have overlapping patterns of expression within the cochlear epithelium, and genetic inactivation of these genes also leads to moderately increased numbers of HCs [15,50,62]. A graded increase in the numbers of HCs is induced by the loss of multiple members of the Hes family, but even in triple mutants, the observed phenotypic defects are not substantial [49]. Intriguingly, Sox9 upregulates Hey1 and HeyL proteins that are known to be direct targets of Notch signaling [50]. The co-existence of two pathways for Hes/Hey factor activation during cochlear differentiation allows cells to respond appropriately to specific environmental cues and ensures robustness in regulating cell fate decisions.

Altogether, these multiple ways to inhibit Atoh1 in cells destined for an SC fate likely reflect the biological importance of a tightly controlled expression and activity to ensure correct cell patterning in the cochlear epithelium. The multiplicity of Atoh1 inhibitors could also explain why depleting Sox9 from progenitor cells in vivo was insufficient to induce Atoh1 expression and HC differentiation.

Hes/Hey factors could inhibit Atoh1 function by two mechanisms [63]. First, these bHLH proteins could act by competing with Atoh1 for dimerization partners that are essential for Atoh1 function. The activity of Atoh1 relies on its interaction with E proteins to bind E boxes present in the promoter of target genes. Since Hes/Hey factors were also shown to interact with these proteins, the upregulation of Hey1 and HeyL could lead to the sequestration of Atoh1 partner, thereby limiting the number of active Atoh1 transcriptional complexes. Second, Hes/Hey factors could compete with Atoh1 for DNA binding. Indeed, Hey1 and HeyL bind preferentially to E boxes, and their upregulation would result in increased occupancy of Atoh1 target promoters. Moreover, Hes/Hey factors could also exert a direct repressive effect on Atoh1 target genes since they were shown to recruit transcriptional repressors like histone deacetylases.

## 5. Conclusions

Here we report a role for Sox9 during cell patterning in the developing mammalian organ of Corti. These data imply that Sox9 mutation leads to a modification of the sensorineural epithelium by affecting the integrity of the Atoh1 pathway and may contribute to the hearing loss seen in a subset of patients with campomelic dysplasia (OMIN No. 114290). In a recent study, it was observed that heterozygous mice carrying the Y440X mutation in the Sox9 gene, which results in the truncation of the C-terminal transactivation domain of Sox9, exhibited neurosensory deafness [64]. However, it appears to be associated with the endolymphatic sac and the stria vascularis. Whether the development of the organ of Corti is unaffected remains to be demonstrated.

In mammals, genetic and/or environmental insults to the cochlea lead to hearing loss, often due to permanent loss or dysfunction of the organ of Corti. Thus, manipulating Sox9 expression or activity may be an important pathway when designing therapies to treat deafness caused by loss or dysfunction of HCs and/or SCs.

## Figures and Tables

**Figure 1 cells-12-02148-f001:**
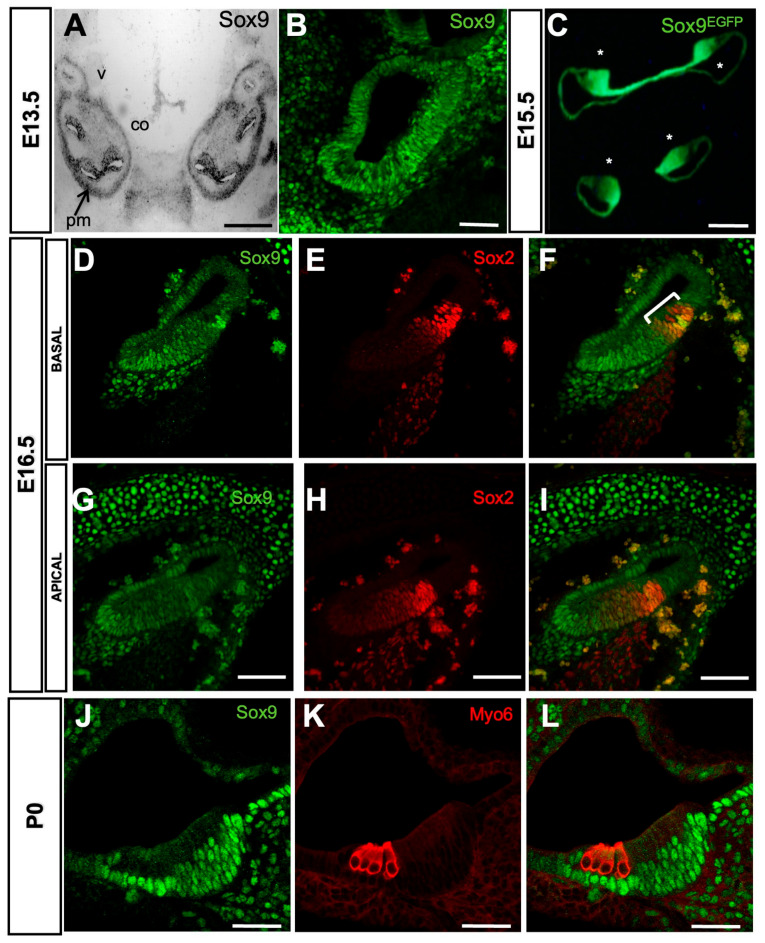
Sox9 expression in the developing cochlea. (**A**,**B**) In situ hybridization (**A**) and immunostainings (**B**) directed against Sox9 in E13.5 mouse inner ear. Sox9 is broadly expressed in the otic capsule and in the cochlear duct epithelium at the transcript and protein levels. (**C**) Cross-section from Sox9EGFP mouse at E15.5 reveals strong Sox9 expression within the cochlear duct (asterisks indicate individual turns of the coiled cochlea). (**D**–**I**) Immunostainings of Sox9 (green) and Sox2 (red) through basal (**D**–**F**) and apical (**G**–**I**) turns of E16.5 cochlea. Sox9 is absent from HCs at the base ((**D**,**F**), differentiating OHCs are indicated by a bracket) but is still detected in future HCs at the apex (**G**,**I**), while Sox2 is present in these cells for both areas (**E**,**F**,**H**,**I**). Sox9 is highly expressed in the surrounding mesenchyme. (**J**–**L**) Representative cross-section through the mid-basal turn of P0 cochlea labeled for Sox9 (green) and Myo6 (red). Sox9 is absent from Myo6-positive HCs and highly expressed in the neighboring SCs. Scale bars: (**A**,**C**) 100 µm, (**B**,**D**–**L**) 50 µm.

**Figure 2 cells-12-02148-f002:**
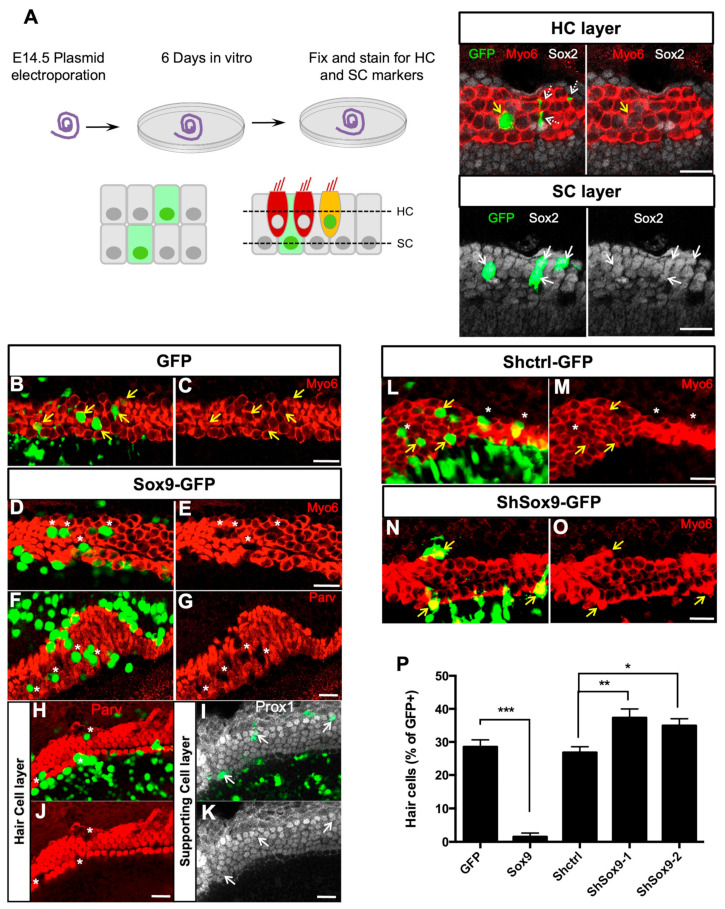
Modulation of Sox9 expression affects HC differentiation. (**A**) Experimental design: E14.5 cochleae were electroporated with GFP-expressing plasmids and further cultured for 6 DIV before fixation and immunohistochemical analysis of HC and SC markers. The total number of GFP-positive cells was evaluated throughout the depth of the sensory epithelium, in the HC and SC layers. In this representative result, only 1/5 GFP+ transfected cell has developed into a Myo6+ HC (indicated by a yellow arrow) and 4/5 have adopted an SC fate (white arrows). Cochlear explants were electroporated at E14.5 with GFP (**B**,**C**) or Sox9-GFP (**D**–**G**) and further cultured for 6 DIV. Yellow arrows = Transfected cells in the sensory epithelium. Cells overexpressing Sox9 within the HC layer rarely express Myo6 (**E**) or Parv (**G**), indicating that they do not develop as HCs (white asterisks). (**H**–**K**) Confocal images taken at the level of HCs (**H**,**J**) or SCs (**I**–**K**) after electroporating Sox9-GFP (white asteriks = transfected cells within the sensory epithelium). While Sox9 overexpression prevents the formation of GFP+Parv+ HCs, it does not affect the number of SCs (GFP+Prox1+) beneath the sensory cells (indicated by white arrows). (**L**–**O**) E13.5 cochleae electroporated with Shctrl (**L**–**M**) or ShSox9-GFP (**N**–**O**) and cultured for 6DIV. Sox9 knockdown increased the population of GFP+Myo6+ HCs (yellow arrows) at the expense of those developing as SCs (white asterisks). (**P**) Quantification of (**A**–**N**) the percentage of Parv+ or Myo6+ cells among the GFP+ cell population was analyzed within the sensory region of each explant. Data are expressed as mean ± SEM (6–10 cochlear explants per condition from 3 independent experiments). * = *p* < 0.05, ** = *p* < 0.01 and *** = *p* < 0.001. Scale bars (**A**–**N**): 25 µm.

**Figure 3 cells-12-02148-f003:**
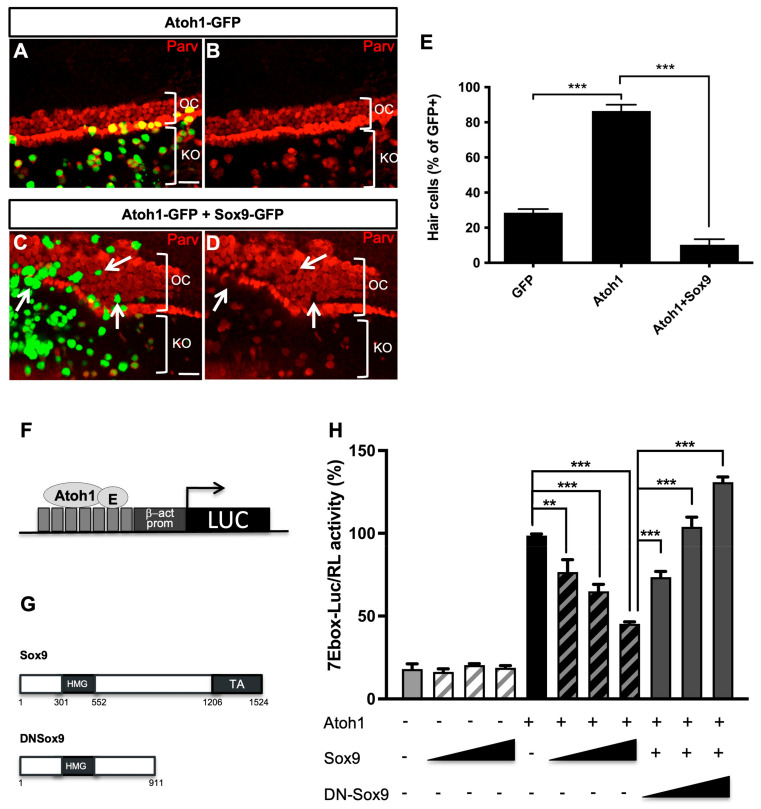
Sox9 inhibits Atoh1 activity. (**A**–**D**) Parvalbumin (Parv) immunostainings of cochlear explants cultured for 6 days after electroporation with Atoh1-GFP alone (**A**,**B**) or in combination with Sox9-GFP (C-D). While most of the cells expressing Atoh1-GFP develop as Parv+ HCs, even in Kölliker’s organ (KO), outside the organ of Corti (OC), most cells expressing Atoh1 together with Sox9-GFP fail to express HC marker. Scale bars (**A**–**D**): 25 µm (**E**) Quantifications of (**A**–**D**). The percentage of GFP+Parv+ cells, within the organ of Corti, was evaluated for each explant. Data are presented as mean ± SEM (6–12 cochlear explants per condition from 3 independent experiments). (**F,G**) Schematic representation of luciferase reporter controlled by Atoh1 and Sox9 constructs (**H**) Luciferase assays of UB/OC1 cells transfected with Atoh1-responsive reporter (7Ebox-Luc) and control GFP, Atoh1-GFP, Sox9-GFP, or DNSox9-GFP expressing vectors. The firefly luciferase activity of the reporter constructs in transfected cells was normalized to the Renilla luciferase activity of the control construct (pRL-SV). Data are presented as a percentage of Atoh1 relative activity from three independent experiments. Atoh1 efficiently activates transcription from E-box-containing promoters, but this is dose-dependently inhibited by the presence of Sox9. The use of a dominant negative form of Sox9 protein (DN-Sox9), lacking the transactivation domain, suppresses Sox9 effect on Atoh1. ** = *p* < 0.01 and *** = *p* < 0.001.

**Figure 4 cells-12-02148-f004:**
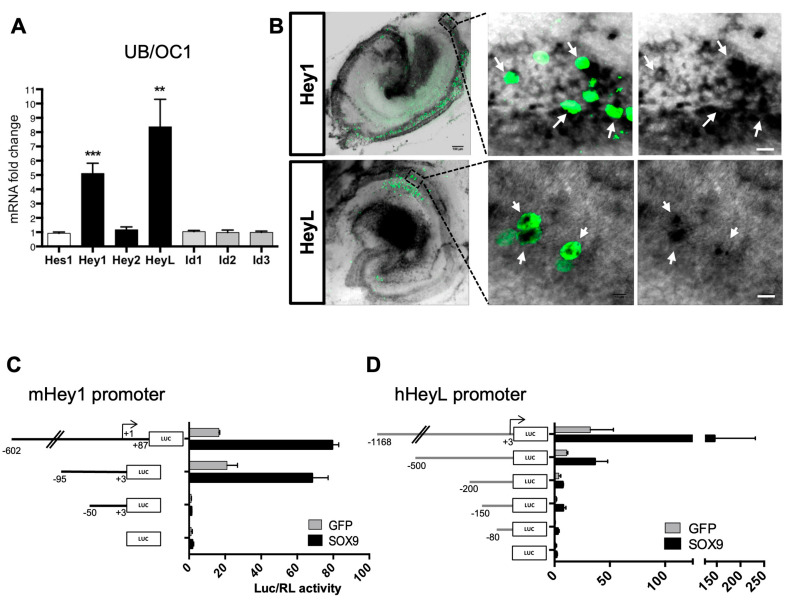
Sox9 upregulates Hey1 and HeyL factors. (**A**) RT-qPCR was performed in UB/OC1 cells transfected with control-GFP or Sox9-GFP plasmids. The expression level was normalized over GAPDH transcript level and reported to control-GFP condition (mean ± SD from 3 individual experiments). (**B**) E14.5 cochleae were electroporated with Sox9-GFP and cultured for 6 DIV. Hey1 and HeyL in situ hybridization were performed before anti-GFP staining. Scale bars left panels: 100 µm, right panels: 10 µm. White arrows indicated transfected cells (**C**,**D**) Luciferase reporter vectors containing fragments from Hey1 (**C**) or HeyL (**D**) promoters as well as controlling empty pLUC4 vectors were transfected in UB/OC1 cells with Renilla control (RL) vector and either GFP or Sox9-GFP plasmid. Data are presented as the normalized Luc/RL ratio reported to control conditions (mean ± SEM from 3 individual experiments). ** = *p* < 0.01 and *** = *p* < 0.001.

**Figure 5 cells-12-02148-f005:**
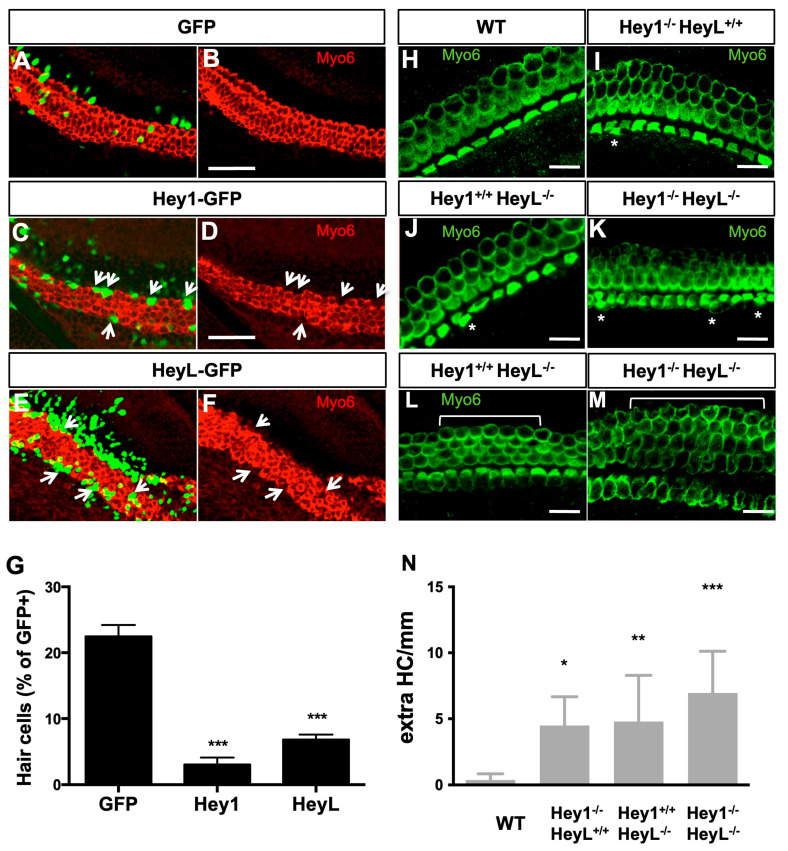
Hey1 and HeyL inhibit hair cell fate. (**A**–**F**) Myo6 immunostainings after electroporating GFP (**A**,**B**), Hey1-GFP (**C**,**D**), or HeyL-GFP (**E**,**F**) in E14.5 cochleae and culturing them for 6 days. Transfected cells overexpressing Hey factors often lack Myo6 staining (arrows), indicating that HC differentiation is blocked by Hey1 and HeyL. (**G**) Quantifications of (**A**–**F**). The percentage of transfected cells developing as HCs was evaluated for each explant. Data are presented as mean ± SEM (6–12 cochlear explants per condition from 3 independent experiments). (**H**–**M**) Myo6 immunolabeling of E18.5 whole-mounted cochleae from wild-type (**H**), Hey1^−/−^ (**I**), HeyL^−/−^ (**J**,**L**), or double knock-out animals (**K**,**M**). In the absence of Hey factors, extra HCs are formed in the organ of Corti (indicated by asterisks for IHC and bracket for OHC). (**N**) Quantifications of supernumerary HCs. The number of extra HCs per mm of cochlea was evaluated for wild-type animals (*n* = 10) or mice lacking expression from Hey1 (*n* = 5), HeyL (*n* = 4), or both genes (*n* = 4). Data are presented as mean ± SEM. * = *p* < 0.05, ** = *p* < 0.01 and *** = *p* < 0.001. The scale bar is 50 µm in (**A**–**F**) and 25 µm in (**G**–**J**).

**Figure 6 cells-12-02148-f006:**
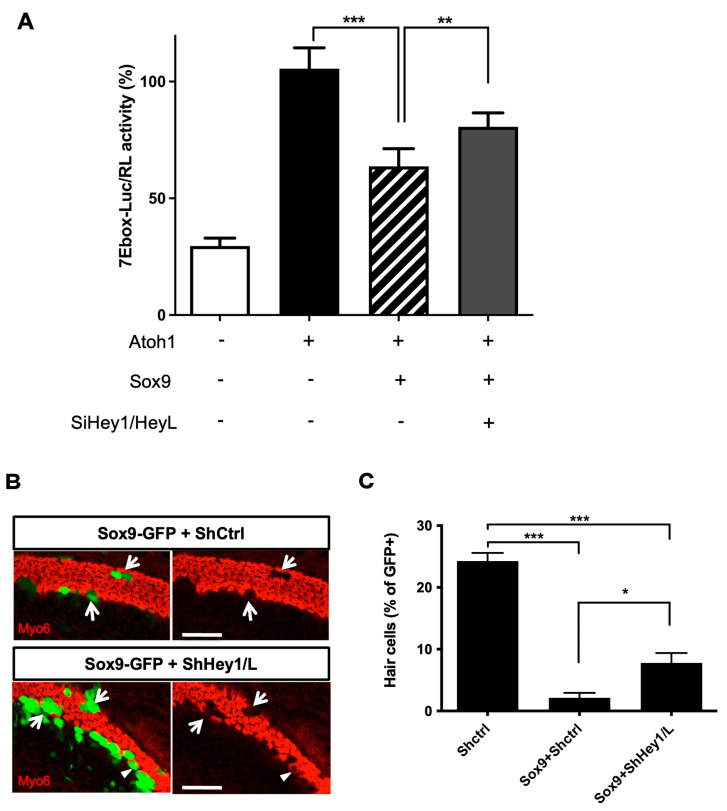
Hey1 and HeyL participate in Sox9 inhibition of Atoh1 activity and inhibition of hair cell fate. (**A**) Luciferase assays of UB/OC1 cells transfected with Atoh1-responsive reporter (7Ebox-Luc), Sox9-GFP vector and siRNA control or directed against Hey1 and HeyL. Data are presented as percentages of Atoh1 relative activity from three independent experiments. When Hey1 and/or HeyL upregulation is prevented by the use of specific siRNAs, Sox9 effect on Atoh1 function is abolished. (**B**) E14.5 cochleae were electroporated with Sox9-expression plasmid together with shRNA-control or shRNA-Hey1/HeyL and further cultured for 6 DIV. While none of the Sox9-transfected cells express Myo6 HC marker (arrows), some of the cells transfected with Sox9 in combination with shHey1 and shHeyL were able to fully differentiate into HCs (arrowhead). (**C**) The percentage of transfected cells (GFP+) that differentiate into hair cell is presented as the mean ± SEM for 8 individual explants within 3 experiments. Scale bar is 50 µm. * = *p* < 0.05, ** = *p* < 0.01 and *** = *p* < 0.001.

**Figure 7 cells-12-02148-f007:**
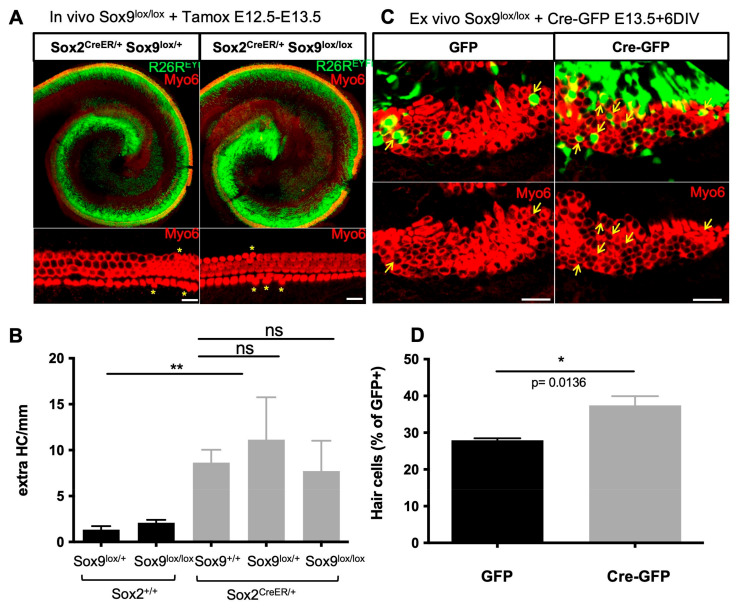
Sox9 loss in prosensory cells leads to some extra HCs ex vivo but not in Sox2CreERT2/+ Sox9Lox/Lox mice in vivo. (**A**) Myo6 immunostainings of whole-mounted P0 cochlea from Sox2CreERT2/+; R26REYFP; Sox9lox/+ (Het) or Sox2CreERT2/+; R26REYFP; Sox9lox/lox (KO) mice injected with tamoxifen at day E12.5 and E13.5. Supernumerary HCs (asterisks) were detected in both genotypes regardless of Sox9 gene status. (**B**) The number of extra HCs was quantified along the entire length of the cochlea and normalized to the cochlear length for all littermates, whether they express the inducible Cre from the Sox2 allele or not (mean ± SD, *n* = 3–6 animals per genotype). Sox2 hypomorphism induces extra HCs, which is not further increased by Sox9 invalidation. (**C**) Myo6 immunostainings after electroporating GFP or Cre-GFP in E13.5 Sox9Lox/Lox cochleae and further cultured for 6 days. An increased proportion of GFP+ cells developing into HCs (arrows) was seen in Cre-transfected cochleae, indicating that HC differentiation is promoted by Sox9 suppression. (**D**) Quantifications of the percentage of GFP+ cells developing as HCs. Data are presented as mean ± SEM (5–8 cochlear explants per condition from 2 independent experiments). * = *p* < 0.05 and ** = *p* < 0.01. Scale bars: 25 µm.

## Data Availability

Not applicable.

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
