# Peer review of "Sox9 Inhibits Cochlear Hair Cell Fate by Upregulating Hey1 and HeyL Antagonists of Atoh1"

_cells, 2023, doi:10.3390/cells12172148_

Round 1
Reviewer 1 Report
Veithen et al. study the effect of Sox9 overexpression in cochlear hair cell development in the in vitro condition. They showed that Hey1 and HeyL might be the key, but not the exclusive targets of the Sox9. Nonetheless, they contribute to the repressive effects of Sox9 on the transcriptional activity of Atoh1. The study did show the effects of Sox9 overexpression on HC development. Nonetheless, because loss of Sox9 did not reveal phenotypes, it clearly weakens the study of Sox9 in this study. The author speculated the redundancy between Sox9 and Sox2, but it was not confirmed. Overal, the study provides useful information to the hair cell development field, albeit it did not bring significant new biological insights.
One main issue: the figures could be improved to clearly visualize the colocalization of GFP and Myo6 in the transfected cells. Current images are not sufficient to precisely appreciate it. It is suggested to use some nuclei marker (such as Insm1 and Bcl11b) to confirm it. Myo6 is a membrane marker and it can be an artificial overlap between two neighboring cells.
The paper is reasonably clear to read, but many typos and a few examples were listed.
1) line 81 ''. '' was missed after ''differentiation''.
2) line 293 ''electroporating these shRNAs in E13.5 cochlear explants '', it is not consistent with the E14.5 listed in Figure 2.
3) line 426 ''25µm in (G-J) '', the (G-J)'' should be'' (H-M) ''
it can be improved.
Author Response
Indeed, our identification of the transfected cell fate is based on GFP co-localization with Myo6, which is a cytoplasmic marker for hair cells. Although we agree that the use of nuclei markers such as Insm1 and Bcl11b would confirm the hair cell fate, it should be noted that both factors are only expressed in outer hair cells and are thus missing from the inner hair cells [1]. With these markers, the total pool of sensory cells would not be considered. Moreover, GFP expressed from our bi-cistronic IRES vector is present in both nucleus and cytoplasm. By analyzing the different focal planes of our confocal images, we think it is more convenient to observe colocalization with a cytoplasmic marker, as most of the cell surface and borders may be observed and assessed for GFP and Myo6 overlap.
To help the reader appreciate the fate of transfected cells we provide orthogonal projections in Supplemental Figure 1.
reference
Wiwatpanit, T.; Lorenzen, S.M.; Cantu, J.A.; Foo, C.Z.; Hogan, A.K.; Marquez, F.; Clancy, J.C.; Schipma, M.J.; Cheatham, M.A.; Duggan, A.; et al. Trans-differentiation of outer hair cells into inner hair cells in the absence of INSM1. Nature 2018, 563, 691-695, doi:10.1038/s41586-018-0570-8.
Reviewer 2 Report
Summary
The authors have demonstrated exemplary proficiency in their analysis of Sox9's pivotal role in hair cell development within the murine cochlea. Their exploration of the underlying mechanisms, culminating in the identification of Atoh1, Hey1, and HeyL as central figures, significantly enhances our understanding of this complex process. Undoubtedly, this compilation of insights holds substantial value for the inner ear community. Nevertheless, while the content itself is noteworthy, there are some apprehensions regarding the presentation of the data that warrant consideration. After revising these points, I believe it would be suitable for publication at Cells.
Major comments
Enhancements in image quality are imperative, as the current fluorescent images suffer from background signal and/or overexposure. Take, for instance, Figure 1D-E, where discerning co-positivity of Sox2 and Sox9 in all Sox2-positive cells becomes arduous due to these technical shortcomings. Unfortunately, these issues extend to impact conclusions. Line 268-269 asserts Sox2's absence, but a meticulous review of Figure S1E-H reveals faint Sox2 signals within several GFP-positive cells, contradicting the claim. Rectifying these concerns is essential for upholding the study's accuracy and integrity.
The methodology employed for quantification, as illustrated in Figure 2O, lacks transparency. The specifics surrounding whether the quantification was manual or automated, the number of observers involved, the criteria for identifying cells as GFP-positive or Myo6-positive, and the inclusion criteria for “ectopic” HCs remain undisclosed within the current Materials and Methods section. Moreover, enhancing clarity for the reader would involve highlighting all counted cells within an image. For instance, in Figure 3C-D, only three arrows are apparent, yet the quantification suggests a substantial 80% decrease, necessitating the identification of all counted cells for a comprehensive understanding. This similar situation is observed in Figure 5 as well. Addressing these issues is paramount for enhancing the rigor and comprehensibility of the study's quantitative assessments.
Certain conclusions drawn lack sufficient support from the provided data:
Line 271-272: the development of SCs is not being evaluated, only their presence or absence.
Line 278: “Cells do not express Myo6…” This is not true as stated before and also visible in the Fig. 2O around 1.5% of the cells do.
Line 312: It’s not shown that Sox9 negatively affects Atoh1 function by these experiments, only that the Atoh1 induced HC occurrence can be counteracted by cotreatment with Sox9.
Line 361: expression profiles are not completely consistent with Sox9 as Sox9 is being expressed throughout the epithelium with loss in the arising hair cells as showed earlier in the manuscript.
Line 370: There is no analysis of the increased expression of Hey1 and HeyL levels performed.
Line 408: Their contribution to otic maturation or terminal SC differentiation is not shown here. Only that hair cell quantity is being increased by deletion and decreased by overexpression.
Line 580-583: The authors show a role of Sox9 in development however a role of HCs in regeneration is not being investigated. So this section seems a bit far stretched.
Minor comments
Line 34-42: References are lacking. Additionally, I would propose to change …the otocyst is divided in two… (line 38) to …the otocysts can be divided in two…
Line 44-47: References are lacking.
Line 66: Upon activation… Activation of what?
Line 81: …differentiation. Gain…
Fig. 1. 1C: Based on the current presentation I cannot judge Sox9-GFP is expressed in the entire epithelium as stated and would therefore recommend a close-up image similar to 1B. I would suggest to switch D-F with G-I as the basal cells are presented first.
Fig. S2: text covered by figures.
Fig. 2: The schematic is actually subfigure A.
Line 318: The UB/OC1 line is already mentioned earlier in the paper, so it should be introduced in this first section.
Line 517: reference lacking.
Line 575-579: this information would be better fitted in the discussion instead of conclusion.
Author Response
Summary
The authors have demonstrated exemplary proficiency in their analysis of Sox9's pivotal role in hair cell development within the murine cochlea. Their exploration of the underlying mechanisms, culminating in the identification of Atoh1, Hey1, and HeyL as central figures, significantly enhances our understanding of this complex process. Undoubtedly, this compilation of insights holds substantial value for the inner ear community. Nevertheless, while the content itself is noteworthy, there are some apprehensions regarding the presentation of the data that warrant consideration. After revising these points, I believe it would be suitable for publication at Cells.
Major comments
Enhancements in image quality are imperative, as the current fluorescent images suffer from background signal and/or overexposure. Take, for instance, Figure 1D-E, where discerning co-positivity of Sox2 and Sox9 in all Sox2-positive cells becomes arduous due to these technical shortcomings.
New figure 1D-L is now provided in the revised version. We took the original images and at a slight lower magnification. Sox9 is nicely present in the surrounding cartilaginous mesenchyme.
Line 268-269 asserts Sox2's absence, but a meticulous review of Figure S1E-H reveals faint Sox2 signals within several GFP-positive cells, contradicting the claim. Rectifying these concerns is essential for upholding the study's accuracy and integrity.
We are aware of the occurrence of Myo6-negative/GFP-positive cells that display faint Sox2 staining, as this was mentioned initially in the main text at lines 266-268 (and is annotated in Figure S1E-H by white arrows):
“Within the HC layer, some of these Myo6-negative Sox9 transfected cells still showed weak staining for Sox2 (Supplemental Figure 1E-H, white arrows), similar to that observed in the neighbouring non-transfected HCs. However, for most of them, Sox2 was absent from the nucleus (Supplemental Figure 1E-H, yellow arrowheads).”
The methodology employed for quantification, as illustrated in Figure 2O, lacks transparency. The specifics surrounding whether the quantification was manual or automated, the number of observers involved, the criteria for identifying cells as GFP-positive or Myo6-positive, and the inclusion criteria for “ectopic” HCs remain undisclosed within the current Materials and Methods section.
The quantification method is now present in the materials and method section.
Moreover, enhancing clarity for the reader would involve highlighting all counted cells within an image. For instance, in Figure 3C-D, only three arrows are apparent, yet the quantification suggests a substantial 80% decrease, necessitating the identification of all counted cells for a comprehensive understanding. This similar situation is observed in Figure 5 as well. Addressing these issues is paramount for enhancing the rigor and comprehensibility of the study's quantitative assessments.
The quantifications presented in Figure 3E do not include the ectopic HCs present in the Kölliker’s organ but only relate to the GFP-positive cells within the organ of Corti (encompassing HC and underlying SC that display strong Sox2 staining). On the representative image of Figure 3A-B, the organ of Corti contains only 13 Atoh1-GFP transfected cells, out of which 11 express Parvalbumin, the HC marker. Overall, 255 transfected cells were analyzed for this experiment (see supplemental table 1). Upon combination of Atoh1and Sox9 plasmids in Figure 3C-D, 17 GFP-positive cells can be observed in the organ of Corti, amongst which only 1 is faintly positive for Parvalbumin. For this group, a total of 386 transfected cells were analyzed within the organ of Corti (see supplemental table 1).
In addition to a precise methodology of HC fate determination that is now included in the material and methods section, we have added details in the text describing the results from Figure 3. We believe these precisions allow for a better understanding of the study.
Certain conclusions drawn lack sufficient support from the provided data:
Line 271-272: the development of SCs is not being evaluated, only their presence or absence.
We have modified this sentence.
Line 278: “Cells do not express Myo6…” This is not true as stated before and also visible in the Fig. 2O around 1.5% of the cells do.
This has been changed for “Cells rarely express Myo6”.
Line 312: It’s not shown that Sox9 negatively affects Atoh1 function by these experiments, only that the Atoh1 induced HC occurrence can be counteracted by cotreatment with Sox9.
We have modified this sentence accordingly.
Line 361: expression profiles are not completely consistent with Sox9 as Sox9 is being expressed throughout the epithelium with loss in the arising hair cells as showed earlier in the manuscript.
Hey 1 and HeyL ISH (as shown in suppl fig 3) showed a different pattern of expression (with HeyL not present at E14.5). However, both transcription factors mRNAs are present in supporting cells and absent in HCs at E16.5 and P1. Therefore, we conclude that “Hey1 and HeyL expression profiles were consistent with the Sox9 expression pattern and Atoh1 modulation in the developing organ of Corti”.
Line 370: There is no analysis of the increased expression of Hey1 and HeyL levels performed.
We have analyzed a minimum of 3 electroporated explants for each factor and our conclusions are drawn from the observation of a stronger staining in Sox9-electroporated cells compared to the neighboring cells. Quantifications of the signal was hazardous because it is difficult to define the exact borders of the epithelial cells present in the organ of Corti. While we could normalize the staining intensity to the cell area for GFP-electroporated cells (based on the surface of GFP staining), it was very tricky to do the same with the surrounding non-electroporated cells. Besides, we do not think that comparing Hey signals between Sox9-GFP and control GFP-electroporated explants would be reliable because of the variability occurring between samples of situ hybridization assays.
Line 408: Their contribution to otic maturation or terminal SC differentiation is not shown here. Only that hair cell quantity is being increased by deletion and decreased by overexpression.
This sentence was modified accordingly.
Line 580-583: The authors show a role of Sox9 in development however a role of HCs in regeneration is not being investigated. So this section seems a bit far stretched.
This part has been removed.
Minor comments
Line 34-42: References are lacking. Additionally, I would propose to change …the otocyst is divided in two… (line 38) to …the otocysts can be divided in two…
A review reference has been added and the edit proposition was followed.
Line 44-47: References are lacking.
References have been added.
Line 66: Upon activation… Activation of what?
Precision has been added.
Line 81: …differentiation. Gain…
The final dot was added to this sentence.
Fig. 1. 1C: Based on the current presentation I cannot judge Sox9-GFP is expressed in the entire epithelium as stated and would therefore recommend a close-up image similar to 1B. I would suggest to switch D-F with G-I as the basal cells are presented first.
We changed figure 1C to show a closer image.
We switched the panels accordingly.
Fig. S2: text covered by figures.
We move the text accordingly.
Fig. 2: The schematic is actually subfigure A.
Line 318: The UB/OC1 line is already mentioned earlier in the paper, so it should be introduced in this first section.
The description of the UB/OC1 cell line is now mentioned earlier.
Line 517: reference lacking.
References have been added.
Line 575-579: this information would be better fitted in the discussion instead of conclusion.
These last lines of the paper are difficult to move into the discussion. We maintained them as they are but can suppress them from the manuscript if needed.